# Comprehensive Analysis of Drug Utilization Patterns, Gender Disparities, Lifestyle Influences, and Genetic Factors: Insights from Elderly Cohort Using g-Nomic^®^ Software

**DOI:** 10.3390/ph17050565

**Published:** 2024-04-28

**Authors:** Bárbara Rodríguez Castillo, Marc Cendrós, Carlos J. Ciudad, Ana Sabater

**Affiliations:** 1Department of Biochemistry and Physiology, School of Pharmacy and Food Sciences, Universidad de Barcelona, 08007 Barcelona, Spaincciudad@ub.edu (C.J.C.); 2Technical Department, Eugenomic, 08029 Barcelona, Spain

**Keywords:** pharmacogenetics, drug adverse reactions, polypharmacy, gender, cholesterol, antihypertensive, platelet aggregation, gastroprotective, lifestyle, g-Nomic^®^

## Abstract

Polypharmacy is a global healthcare concern, especially among the elderly, leading to drug interactions and adverse reactions, which are significant causes of death in developed nations. However, the integration of pharmacogenetics can help mitigate these risks. In this study, the data from 483 patients, primarily elderly and polymedicated, were analyzed using Eugenomic^®^’s personalized prescription software, g-Nomic^®^. The most prescribed drug classes included antihypertensives, platelet aggregation inhibitors, cholesterol-lowering drugs, and gastroprotective medications. Drug–lifestyle interactions primarily involved inhibitions but also included inductions. Interactions were analyzed considering gender. Significant genetic variants identified in the study encompassed *ABCB1, SLCO1B1, CYP2C19, CYP2C9, CYP2D6, CYP3A4, ABCG2, NAT2, SLC22A1*, and *G6PD*. To prevent adverse reactions and enhance medication effectiveness, it is strongly recommended to consider pharmacogenetics testing. This approach shows great promise in optimizing medication regimens and ultimately improving patient outcomes.

## 1. Introduction

The definition of “polypharmacy” or “polymedicated patient” remains a subject of debate, with varying interpretations. While the World Health Organization (WHO) broadly defines it as the administration of multiple medications simultaneously, the most commonly used definition involves the use of five or more medications daily [1,2]. Polypharmacy is a prevalent issue among elderly individuals with multimorbidity, which refers to the coexistence of two or more chronic health conditions. It poses significant challenges in healthcare settings worldwide, leading to medication non-adherence, increased risk of drug duplication, drug–drug interactions, adverse drug reactions (ADRs), and heightened healthcare costs. Drug–drug interactions occur when one drug, known as the precipitating drug, modifies the pharmacological or pharmacokinetic properties of another drug, known as the object drug [1,2,3,4].

ADRs are a leading cause of mortality in developed countries, with over 100,000 deaths reported annually in the United States alone, even among individuals adhering to prescribed medication protocols. ADRs are more common among the elderly population and can be attributed to a combination of factors such as reduced organ function, comorbidities, polypharmacy, age-related pharmacokinetic changes, and pharmacodynamics variations. Integration of pharmacogenetics criteria could potentially anticipate and prevent 30% to 60% of ADRs [5,6,7].

Pharmacogenetics focuses on how variations in single genes affect an individual’s response to a specific drug [8,9,10].

When a drug is administered, it undergoes processes of absorption, distribution, metabolism, and excretion (ADME). Pharmacogenetics effects can influence pharmacokinetic (PK) and pharmacodynamics (PD) factors. PK factors involve the absorption, distribution to the site of action, metabolism, and elimination of a drug. Genetic polymorphisms can lead to changes in drug concentrations at the target site [8,9,11]. PD factors encompass the drug’s target and the downstream signaling cascades. Genetic variations in drug targets can result in measurable differences in an individual’s response to a drug, altering the biological or physiological response, and these variations can be associated with one or more specific genes [8,9].

Genomic differences between individuals occur approximately every 300 to 1000 nucleotides, leading to more than 14 million single nucleotide polymorphisms (SNPs) distributed throughout the human genome, contributing to human variability. Pharmacogenetics research has highlighted the significant role played by these genetic variants, particularly in genes encoding drug-metabolizing enzymes, drug transporters, and therapeutic targets, in producing diverse responses to treatment among individuals [5,10,12].

Genetic variations in drug-metabolism-related genes contribute to the observed differences in drug efficacy and the occurrence of ADRs, particularly among the elderly population. Whole-genome mapping techniques, including SNP array testing and next-generation sequencing, have enabled researchers to identify and validate genetic markers associated with serious ADRs. These markers facilitate the screening of patients at risk and the integration of molecular and clinical information, leading to the development of precision medicine approaches. Besides genetic factors such as mutations (including SNPs), gene deletions, and duplications, other factors influencing medication response include physiological conditions (age, gender, body size), environmental influences (diet, lifestyle habits), and pathological factors (liver and renal function, diabetes).

Precision medicine considers all these factors on an individual patient basis and aims to understand the biological basis of diseases, enabling the prescription of customized treatments tailored to each patient’s unique conditions [12,13]. This discipline also plays a crucial role in the development of medications with improved outcomes for patients.

The g-Nomic^®^ software, developed by Eugenomic^®^, was utilized to analyze polypharmacy in this study. This software offers a thorough examination of medications, including their substrates, primary and secondary metabolic pathways, inhibitory and inductive effects (categorized as weak, moderate, or strong according to FDA criteria), drug interactions, interactions with lifestyle habits, associated risks, guideline alerts, and individual genetic information such as single nucleotide polymorphisms (SNPs). When patient genetic data is available, g-Nomic^®^ assists in categorizing individuals as normal metabolizers (NM), intermediate metabolizers (IM), ultra-rapid metabolizers (UM), or poor metabolizers (PM). With access to a patient’s genetic profile, g-Nomic^®^ provides comprehensive pharmacogenetic interpretations, covering over 1100 genetic markers associated with drug response types, including normal response, toxicity, and therapeutic failure. The software encompasses genetic variations of drug-metabolizing enzymes, transmembrane transporter proteins (impacting drug pharmacokinetics), and therapeutic targets (affecting pharmacodynamics). Each gene’s level of evidence is indicated based on established guidelines, technical sheets, and relevant scientific publications. g-Nomic^®^ receives regular updates and reviews based on drug labels from official entities at national and international levels (such as FDA, EMA, and AEMPS) and guidelines from organizations like CPIC, DWPG, and PharmGKB. Periodic reviews with specialized publications ensure the software’s information remains current and accurate.

The main objectives of this study encompass a multifaceted analysis of drug utilization patterns, gender-specific trends, interactions among medications, and the influence of lifestyle habits on drug metabolism in the liver [14]. Firstly, we aimed to determine the most frequently utilized active ingredients of drugs, highlighting categories such as antihypertensive agents, cholesterol-lowering medications, and analgesics. Secondly, we investigated gender-specific utilization patterns, delineating differences in drug preferences between female and male populations. Thirdly, we explored the interplay between medications and lifestyle habits, identifying common active ingredients and potential interactions. Additionally, we reviewed potential interactions among the most frequently used drugs, focusing on their combined effects and possible side effects. Lastly, we looked into the genetic aspects associated with drug metabolism, identifying relevant genes and haplotypes potentially influencing drug responses. Through these comprehensive analyses, our study aims to provide valuable insights into medication utilization patterns, gender disparities, lifestyle influences, and genetic factors affecting drug metabolism within the studied population. 

## 2. Results

### 2.1. Determination of Most Frequent Active Ingredients of Drugs

The patient database contains 329 active drug ingredients, categorized based on their therapeutic effect or treatment type; it is presented in Table 1, which provides the corresponding percentages of the population that utilized at least one drug from each category. The most used drugs and treatments include antihypertensives, platelet aggregation inhibitors and anticoagulants, cholesterol-lowering medications, gastroprotective agents, treatments for sleep disorders, diuretics, and analgesics. Furthermore, a gender-based analysis reveals that the use of analgesics, anxiolytics, thyroid hormone therapy, treatment for osteoporosis, and sleep disorder medications is more prevalent among female patients, while treatments for gout and overactive bladder (OAB) are more frequent among male patients.

Below we present Figure 1 representing the most prevalent active ingredients. Omeprazole was utilized by 39.75% of the population, followed by acetylsalicylic acid (28.36%), atorvastatin (24.84%), bisoprolol (20.08%), simvastatin (18.22%), acetaminophen, and the combination of acetaminophen with tramadol (16.77% and 6.83%, respectively). Levothyroxine ranks seventh in terms of usage, with a prevalence rate of 15.73%, followed by amlodipine (13.66%), metformin (13.04%), hydrochlorothiazide (12.84%), enalapril (11.59%), valsartan (9.73%), lorazepam (9.32%), allopurinol (7.45%), furosemide (7.45%), pantoprazole (7.25%), dipyrone (7.04%), citalopram (6.63%), and olmesartan (6.21%).

### 2.2. Analysis of Gender-Specific Utilization of Active Ingredients

To explore gender perspectives, we investigated the most frequently used active ingredients among the female and male populations. Figure 2 displays the frequency of use by women (blue) and men (red). Omeprazole is the most used active ingredient by both genders, with a prevalence of 40.67% among women and 38.60% among men. Acetylsalicylic acid follows closely, with a utilization rate of 23.51% among females and 34.42% among males.

For the female population, the most common drugs are: levothyroxine (22.39%), simvastatin (21.27%), atorvastatin (20.90%), acetaminophen (19.78%), bisoprolol (15.67%), enalapril (12.69%), hydrochlorothiazide (11.57%), metformin (10.45%), amlodipine (10.07%), lorazepam (10.07%), acetaminophen combined with tramadol (9.70%), valsartan (8.96%), dipyrone (8.96%), furosemide (8.21%), diazepam (8.21%), and chondroitin sulfate (7.46%).

For the male population, the most prevalent drugs are as follows: atorvastatin (29.77%), bisoprolol (25.58%), amlodipine (18.14%), metformin (16.28%), simvastatin (14.42%), hydrochlorothiazide (14.42%), allopurinol (13.49%), acetaminophen (13.02%), tamsulosin (11.63%), valsartan (10.70%), enalapril (10.23%), dutasteride combined with tamsulosin (9.77%), lorazepam (8.37%), acenocoumarol (8.37%), and levothyroxine (7.44%).

### 2.3. Analysis of Most Frequent Active Ingredients in Lifestyle Habits

The role of lifestyle habits in combination with medications is crucial to consider. Among the total population, 96 patients (19.88%) did not report the use of active ingredients related to lifestyle habits, while 148 (30.64%) confirmed the use of between one and four active ingredients. The remaining 239 patients (49.48%) consumed a minimum of five active ingredients from lifestyle habits, with 131 of them (27.12%) using between 10 and 34 ingredients at the time of the study. Lifestyle habit products are substances consumed regularly by patients that are not classified as conventional pharmaceutical drugs but can still interact with medication. These include dietary supplements, food items, and herbal remedies. Active ingredients, on the other hand, are compounds present in both pharmaceutical drugs and lifestyle substances.

Using g-Nomic^®^ personalized prescription software, we identified the active ingredients associated with prescribed lifestyle habits. In Figure 3 it is shown the twenty most frequently consumed lifestyle habit product by the studied cohort. Caffeine was the most common, reported by 31.68% of the population, followed by green tea (26.92%), alcohol (26.50%), grapefruit (24.22%), iron (23.19%), vitamin C (22.15%), calcium (21.53%), vitamin E (21.33%), dairy products (19.05%), pineapple (17.39%), cinnamon (16.98%), antacids (16.77%), chamomile (16.56%), omega 3 (15.53%), magnesium (14.70%), potassium (14.08%), oat bran (13.25%), vitamin D (11.80%), turmeric (11.39%), and ginger (11.18%).

To analyze the gender perspective, we determined the most frequent active ingredients related to lifestyle habits in the female and male populations. Figure 4 displays the frequency of use by women (blue) and men (red). The consumption of most active ingredients related to lifestyle habits is similar between males and females, except for calcium, vitamin D, and calcifediol, which are significantly higher in women compared to men.

For the female population, the most frequent lifestyle habit products are as follows: calcium (30.97%), caffeine (28.36%), green tea (27.61%), grapefruit (22.39%), alcohol (21.64%), iron (21.64%), vitamin C (20.15%), vitamin E (19.03%), chamomile (19.03%), dairy products (18.21%), pineapple (18.21%), magnesium (16.79%), vitamin D (16.79%), oat bran (16.28%), cinnamon (16.04%), antacids (14.18%), potassium (13.43%), turmeric (12.69%), omega 3 (12.31%), and calcifediol (11.19%).

For males, the most frequent lifestyle habit products are caffeine (35.81%), alcohol (32.56%), grapefruit (26.51%), green tea (26.05%), iron (25.12%), vitamin C (24.65%), vitamin E (24.19%), dairy products (20.00%), antacids (20.00%), omega 3 (19.53%), cinnamon (18.14%), pineapple (16.28%), potassium (14.88%), chamomile (13.49%), magnesium (12.09%), ginger (11.63%), calcium (9.77%), turmeric (9.77%) oat bran (10.82%), and vitamin B12 (6.72%).

### 2.4. Interactions among the Most Frequent Drugs

To explore potential interactions between the 20 most frequent drugs (listed in Section 2.1), we utilized the g-Nomic^®^ software, which generated a comprehensive analysis of reported interactions from the existing literature. Table 2 presents the potential side effects that may arise from combining these drugs. The “Objective Drugs” refer to the commonly used drugs in the study population, while the “Precipitating Drugs” are those that can interact and influence the metabolism of the objective drugs.

Upon reviewing Table 2, it becomes apparent that acetaminophen, allopurinol, and hydrochlorothiazide do not appear in the list of drug active ingredients. This is because their metabolism does not primarily rely on pathways affected by enzyme polymorphisms or transporter proteins, thus not generating g-Nomic^®^ messages. However, the “interactions” section of the software provides information indicating that hydrochlorothiazide may enhance allopurinol toxicity in some patients. Although the exact causal relationship or mechanism of interaction has not been established, it is suspected to be associated with decreased renal function.

According to the FDA and visualized in g-Nomic^®^, weak inhibitors may increase the bioavailability of drugs by a factor ranging from 1.20 to 1.9. Moderate inhibitors may increase the bioavailability by a factor of 2.0 to 4.9, while potent inhibitors may increase it by a factor of more than 5. Molecular inhibitors pose a risk of overdose toxicity. The software suggests that dose adjustments or modifications to the treatment regimen may be necessary in such cases [15].

Furthermore, there are weak inducers that can accelerate the metabolism of other drugs, potentially leading to treatment failure unless appropriate dose adjustments or modifications are made. According to the software, patients may require higher doses of the drug, typically an increase of 20% to 30%, to achieve the desired therapeutic effect.

### 2.5. Interactions between the Most Frequent Drugs and Lifestyle Habit Products

To investigate potential interactions between the 20 most frequent drugs (as mentioned in Section 2.1) and the active ingredients of lifestyle habit products (as described in Section 2.2), we used the g-Nomic^®^ software. This analysis generated a comprehensive assessment of reported interactions from the existing literature and Table 3 presents the possible side effects resulting from these combinations.

Like the previous section, lifestyle habit products can act as weak, moderate, or potent inhibitors, thereby increasing the bioavailability of drugs and potentially posing an overdose risk. It is crucial to consider these interactions between drugs and lifestyle habits as they can have a significant impact on treatment outcomes. The g-Nomic^®^ software provides valuable insights and recommendations of the potential risks and adjustments necessary to optimize patient care and therapeutic efficacy, when available.

### 2.6. Genes and Haplotypes Associated with the Metabolism of the Most Frequent Drugs in the Study Population

g-Nomic^®^ software was utilized to analyze the most frequent drugs and lifestyle habits, with a resulting comprehensive report with the genes associated with the metabolism of each drug, along with their corresponding level of evidence. This evidence level was determined based on clinical annotations, providing a score that reflects the overall strength of the underlying evidence. The evidence level classification according to g-Nomic^®^ is as follows: 

1. Recommended: The marker is validated and recommended for guiding therapy, with specific instructions available in technical data sheets or guides. 2. Validated: Good quality information links the marker to relevant clinical effects, making it beneficial for therapy guidance. However, exact doses and contraindications require further assessment. 3. Actionable: Good quality data link the marker to plasma levels, aiding in treatment adjustment. However, evidence of clinical consequences is lacking, making it more useful when the patient’s genotype is known. 4. Informative: The marker is important in drug metabolism, but data linking it to efficacy, toxicity, or plasma levels are insufficient. It may have limited or inconclusive data, and lack of consensus renders it less relevant.

Table 4 presents the genes linked to the metabolism of the following drugs: acetaminophen, acetylsalicylic acid, allopurinol, amlodipine, atorvastatin, bisoprolol, citalopram, dypirone, enalapril, lorazepam, omeprazole, simvastatin, tramadol, and valsartan. However, after searching the PharmGKB and CPIC databases, it was determined that hydrochlorothiazide, pantoprazole, and furosemide do not possess relevant markers with clinical impact for metabolism genes. The associated genes for hydrochlorothiazide and furosemide had low levels of evidence and lacked clinical relevance. Regarding pantoprazole, the level of evidence was classified as 1A according to PharmGKB. In CPIC, a guideline for CYP2C19 and proton pump inhibitor dosing was found, recommending pharmacogenomic testing to determine a patient’s phenotype to prevent the risks of therapeutic failure or potential risk of toxicity [16].

### 2.7. Total Number of Concurrent Medications

Among the patients, 15 individuals (3.11%) reported not using any prescription medication, while the majority of the population used between two and ten drugs. Specifically, 131 patients (27.12%) were using between one and four medications, while the remaining 337 patients (69.77%) were classified as polymedicated individuals, consuming a minimum of five drugs. Among the polymedicated patients, 71 individuals (14.70%) were taking between 10 and 23 drugs, as shown in Figure 5. The anonymous given information documented the various medications consumed by each patient, including both commercial and active ingredient names. Additionally, information on lifestyle-related products associated with each patient was included when available.

## 3. Discussion

The study population predominantly used the following antihypertensive drugs—bisoprolol, amlodipine, enalapril, valsartan, and olmesartan—collectively accounting for 61.27% of the prescriptions (Figure 2). A recent nationwide study conducted in Spain revealed a prevalence of 42.6% for hypertension in the population, with a higher prevalence among men (49.9%) compared to women (37.1%). These findings support the results presented in Table 1 and Figure 2.

Platelet aggregation inhibitors/anticoagulants and cholesterol-lowering drugs were also frequently prescribed in the study population. These medications are commonly administered to patients with cardiovascular disease in conjunction with antihypertensives. This observation is not surprising, considering that cardiovascular diseases are the leading cause of mortality in Spain, accounting for 28.3% of deaths in the country in 2018, according to the Instituto Nacional de Estadística [17].

Among the platelet aggregation inhibitors, acetylsalicylic acid (28.36%) was the most used. Acetylsalicylic acid is often combined with cholesterol-lowering drugs, such as statins, to reduce the risk of vascular events and mortality in patients with cardiovascular diseases [18].

Polypharmacy is often associated with adverse drug reactions (ADRs), with gastrointestinal bleeding and cardiovascular complications being the most common. These complications are frequently linked to the use of nonsteroidal anti-inflammatory drugs (NSAIDs) such as dipyrone, platelet aggregation inhibitors, anticoagulants like acetylsalicylic acid, and antihypertensive drugs [19].

To prevent or reduce gastrointestinal complications, gastroprotective drugs, particularly proton pump inhibitors (PPIs) like omeprazole and pantoprazole, are commonly prescribed. In this study, the combined frequency of omeprazole and pantoprazole use was 47%, and a total of 51.17% of the study population took gastroprotective drugs.

When examining the gender perspective of the most prescribed medications (Figure 3), it was observed that female patients exhibited a higher prevalence of analgesics and anti-inflammatory drugs, including acetaminophen, acetaminophen combined with tramadol, and dipyrone. Additionally, female patients showed a greater frequency of anxiolytics and treatments for sleep disorders, such as lorazepam and diazepam, as well as thyroid hormone therapy (levothyroxine) and osteoarthritis treatment (chondroitin sulfate), in comparison to male patients. Conversely, gout and overactive bladder (OAB) treatments were more frequent in men. Sandin Wranker et al. [20] explain that pain is more commonly reported by women, especially in the vertebral column and legs, and is associated with a lower quality of life among elderly women. Moreover, physiological changes that occur after menopause can lead to symptoms related to postmenopausal syndrome, such as insomnia, osteoporotic symptoms, depression, headache, and vasomotor symptoms, among others [21]. These factors may explain the differences in the intake of analgesics, anti-inflammatory drugs, and sleep disorder and anxiety treatments between female and male populations. Additionally, females have a higher incidence of hypothyroidism and hyperthyroidism than males after menopause [22]. OAB is a prevalent condition in both men and women, but men with a history of prostate problems, such as benign prostatic hyperplasia, have a higher prevalence of OAB [23]. The higher intake of prostate treatments among males (representing 37.67% of the male population) may explain the differences in the use of OAB treatment between men and women. As for gout treatment, evidence suggests that women are protected against it due to the effect of female sex hormones, leading to a higher prevalence of gout in men across all age groups [24].

Drug–drug interactions are not the only interactions that can affect the pharmacokinetics of a drug. Interactions can also occur between drugs and lifestyle habit products, such as food–drug interactions and herb–drug interactions, as well as interactions between drugs and other habits like tobacco and substance abuse. Therefore, it is crucial to consider these factors to achieve personalized prescribing [5,25]. Consequently, one of the objectives of this study was to determine the most frequent lifestyle habit products in the study population and identify potential risks of drug–lifestyle habit product interactions.

Clinically relevant drug–drug and lifestyle product–drug interactions primarily affect the bioavailability of the objective drug. Drugs like omeprazole or lifestyle habit products such as green tea, caffeine, and certain fruits like pineapple and grapefruit are enzyme inhibitors. These inhibitors primarily affect enzyme levels by either blocking or competing at the site of metabolism. The types of inhibitors include competitive inhibitors (binding to the active site of the enzyme), uncompetitive inhibitors (binding to the drug–enzyme complex to inhibit), and non-competitive inhibitors (binding to a different site other than the site of metabolism). All these inhibitors can increase the bioavailability of drugs, leading to a risk of overdose and toxicity. On the other hand, inducers act by increasing gene transcription, resulting in higher enzyme content. Some commonly used drugs like dipyrone or herbs like turmeric, as well as alcohol, are considered inducers that can decrease the bioavailability of objective drugs and potentially lead to treatment failure. Both inhibition and induction processes increase the overall metabolic rate and can cause significant alterations in the patient’s health if not considered by the prescribing physician. In the presence of inhibitors, drug exposure increases, requiring a decrease in dose, dosing interval, or both [14,25].

Drug metabolism involves a complex process occurring in three phases. In phase I, drugs are metabolized by the CYP450 superfamily of enzymes, converting them into water-soluble products for excretion. Phase II involves the enzymatic conjugation of drugs or metabolites from phase I with hydrophilic endogenous compounds, facilitated by transferase enzymes such as UDP-glucuronosyltransferases (UGTs) and N-acetyltransferases (NATs). Finally, in phase III, transmembrane proteins known as drug transporters facilitate the transport of molecules across cell membranes for excretion. ATP-binding cassette (ABC) and solute carrier (SLC) transporters are the main proteins involved in phase III pathways [10,14,26].

Considering that most of the study population consists of elderly individuals who take more than five drugs per day, and some even consume between 10 and 23 drugs, it is recommended to perform pharmacogenetic testing to prevent ADRs. The use of a pharmacogenetics interpretation software could be beneficial in this context. g-Nomic^®^ allows the input of an unlimited number of drugs for a patient, providing information on side effects, possible interactions (including interactions with lifestyle habits), and comprehensive reports. The software enables physicians to review the interactive list of drugs and lifestyle habits with the patient, confirming their regular consumption and selecting relevant items. This generates a report that includes the associated side effects of each drug, drug–drug interactions, drug–lifestyle product interactions, and even interactions between different lifestyle habit products (e.g., reduced calcium absorption when combined with caffeine). The report also includes a section dedicated to identifying genes that may affect drug response or metabolism. g-Nomic^®^ can be implemented not only in medical doctors’ offices but also in hospitals, elderly care facilities, and other healthcare settings.

## 4. Materials and Methods

### 4.1. Study Setting and Data Collection

A database of patients undergoing pharmacological treatments was provided by a health resort located in the province of Valencia, Spain. The data was transferred to Eugenomic^®^, a leading company specializing in genomic medicine and pharmacogenetics. To ensure patient confidentiality, all information was anonymized prior to transfer. Each entry in the database was assigned a unique accession number, and any identifiers linking these numbers to patient identities were removed. As a result, the anonymized database contained no identifiable information that could be used to trace back to individual patients.

### 4.2. Study Population

The study population consisted of 483 patients, comprising 215 males (44.51%) and 268 females (55.49%) of the total population, respectively. The age range of the study population spans from 40 to 92 years. 

### 4.3. Data Analysis

The information from the database was manually input into the g-Nomic^®^ personalized prescription software (version 2.7.2723). This software encompasses a comprehensive collection of more than 2400 active ingredients, which includes medications, food items, dietary supplements, natural herbs, and even substances of abuse. Among these active ingredients, 1205 have associated pharmacogenetics information, as either substrates, inducers, or inhibitors of classic pharmacogenetics pathways.

### 4.4. Determination of the Most Frequent Active Ingredients of Drugs

After uploading all patient data to g-Nomic^®^, the software provided the corresponding active ingredient for each commercial name of the drug. Based on this information, the commercial names in the Excel database were replaced with their respective active ingredients. This allowed us to analyze the frequency of general drug use and assess any gender disparities by determining the proportion of men and women using the twenty most prescribed active ingredients.

### 4.5. Determination of the Most Frequent Active Ingredients of Lifestyle Habit Products

Regarding lifestyle habits, it should be noted that, once the drug-related information for each patient was uploaded to g-Nomic^®^, the software automatically generated a dynamic list of possible active ingredients associated with lifestyle habit products that could potentially interact with the prescribed drugs. Physicians asked patients about the use of these active ingredients and, if confirmed, the ingredient was added to the list of consumed lifestyle habit products, to assess the potential side effects of the drug–lifestyle habit product combination. Therefore, every active ingredient associated with a lifestyle habit product appearing in the Excel database was linked to a drug–lifestyle habit product interaction. The information obtained from g-Nomic^®^ was transferred to the Excel database for organization and analysis of the general frequency of use, as well as gender disparities, by determining the proportion of men and women using the twenty most frequent lifestyle habits.

### 4.6. Determination of Molecular Interactions between the Most Frequent Drugs

After identifying the twenty most used drugs, a new record was created in g-Nomic^®^, specifically selecting these drugs. The software includes a “PGX Report” section that provides a comprehensive overview of molecular interactions between the selected drugs. This allowed us to observe all potential side effects associated with specific drug combinations, as reported by g-Nomic^®^.

### 4.7. Determination of Molecular Interactions between the Most Frequent Drugs and Lifestyle Habit Products

Following the determination of molecular interactions between the twenty most frequently used drugs, the twenty most common active ingredients of lifestyle habit products were added to the same record. Once again, the “PGX Report” section was utilized to observe all potential side effects associated with the combinations of drugs and lifestyle habit products.

### 4.8. Genes and Haplotypes Associated with the Metabolism of the Most Frequent Drugs in the Study Population

After determining the molecular interactions between the most commonly used drugs and lifestyle habit products, the associated genes involved in the metabolism of each drug were examined in the “Genes” section of g-Nomic^®^. For drugs without genetic information provided by the software, a thorough literature search was conducted using the Clinical Pharmacogenetics Implementation Consortium (CPIC) and the Pharmacogenomics Knowledgebase (PharmGKB). These freely available web resources provide comprehensive information on how genetic variations influence drug responses and offer detailed gene/drug clinical practice guidelines [8]. The most significant haplotypes were researched in the Very Important Pharmacogene (VIP) summaries of PharmGKB, while allele frequencies were obtained from dbSNP, the NCBI database of genetic variation.

## 5. Conclusions

In conclusion, polypharmacy in the elderly population presents a significant risk for adverse drug reactions (ADRs), which can be mitigated through two main approaches. Firstly, identifying potential drug–drug interactions using comprehensive databases can prevent ADRs at a basic level. Secondly, considering patients’ genetic polymorphisms affecting drug metabolism, and applying pharmacogenetic criteria, can further personalize prescriptions and reduce the risk of ADRs. Cardiovascular disease treatments are the most prescribed in this population.

Both drug–drug and lifestyle product–drug interactions can affect drug metabolism. Enzyme inhibition from these interactions may elevate drug bioavailability, potentially leading to overdose, while enzyme induction may result in therapeutic failure.

Haplotypes and genetic variants related to drug metabolism significantly influence ADRs by altering protein functions involved in drug metabolism, leading to various phenotypes.

The g-Nomic^®^ personalized prescription software integrates a comprehensive database of drug interactions at various levels, including absorption, distribution, metabolism, and excretion. By conducting pharmacogenetic analysis and considering lifestyle habits, g-Nomic^®^ ensures a comprehensive approach to personalized medicine. 

The limitation of the study is that we relied solely on the g-Nomic database for information regarding drug interactions and the interplay between drugs and lifestyle factors, without conducting additional research. The primary outcome of the study is the awareness of the risks of the polymedication administered to the patients, not only because of the drug–drug interactions but also because of their lifestyles and genetic background if known. Recommendations were positive and most likely avoided some ADRs. This study also contributed to the implementation of pharmacogenetics at this elderly resort.

It should be noted that the study’s focus on only the interactions of the 20 most frequently prescribed drugs constitutes a limitation. Although the studied cohort encompassed 330 active ingredients, considering all possible combinations would result in a vast number of permutations, estimated at approximately 54,285 pairs. Consequently, for practical reasons, we narrowed our analysis to the combinations involving the top 20 most prescribed drugs, accounting for their respective frequencies.

Implementing strategies such as utilizing comprehensive drug interaction databases and integrating pharmacogenetics analysis through tools like g-Nomic^®^ can greatly contribute to optimizing medication regimens and reducing the occurrence of ADRs.

### Patents

g-Nomic^®^ is a registered product by Eugenomic^®^.

## Figures and Tables

**Figure 1 pharmaceuticals-17-00565-f001:**
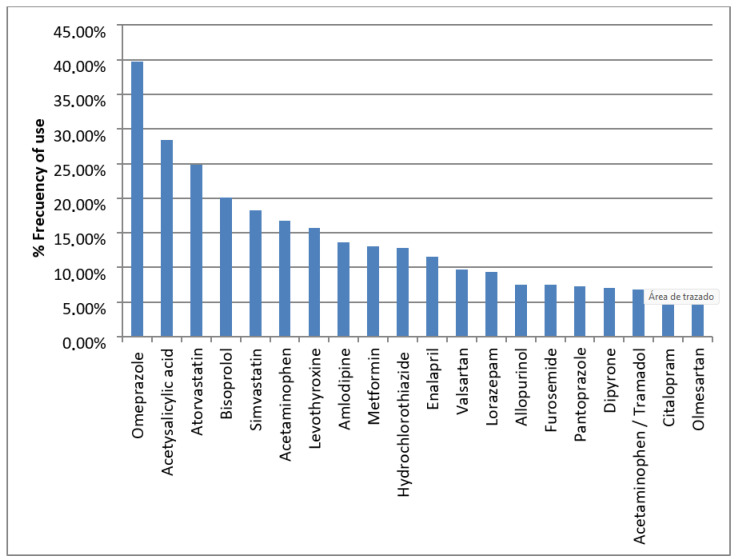
Most frequent active ingredients of the drugs used by the total study population.

**Figure 2 pharmaceuticals-17-00565-f002:**
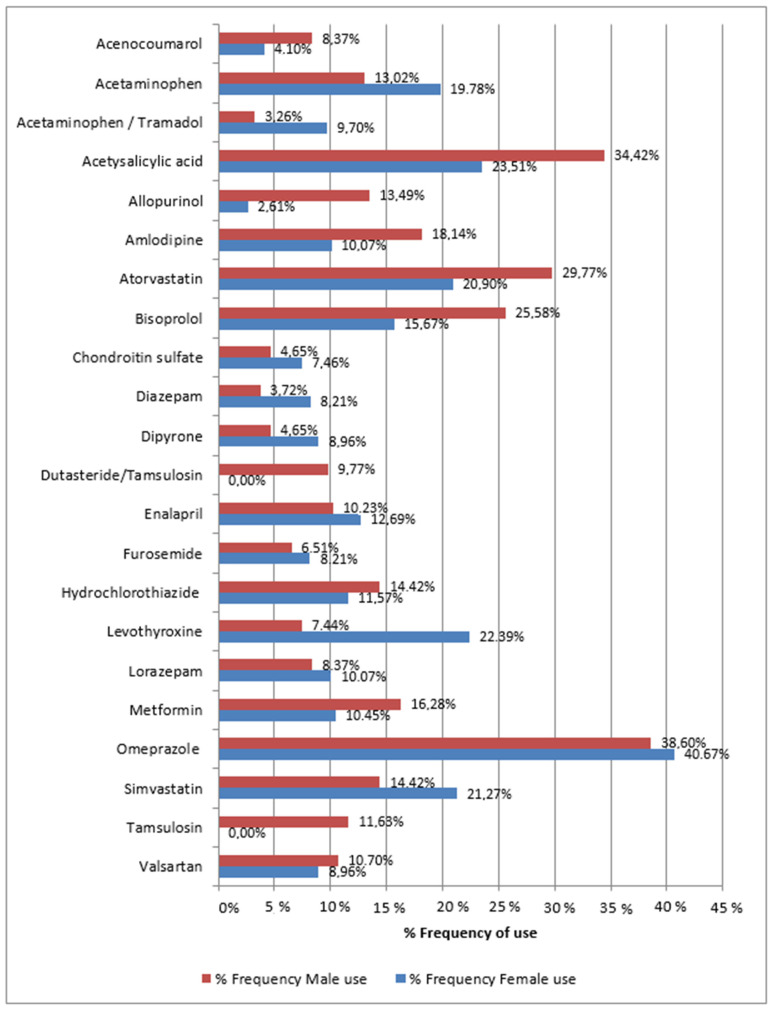
Most frequent drugs active ingredients used by female (blue) and male (red) population.

**Figure 3 pharmaceuticals-17-00565-f003:**
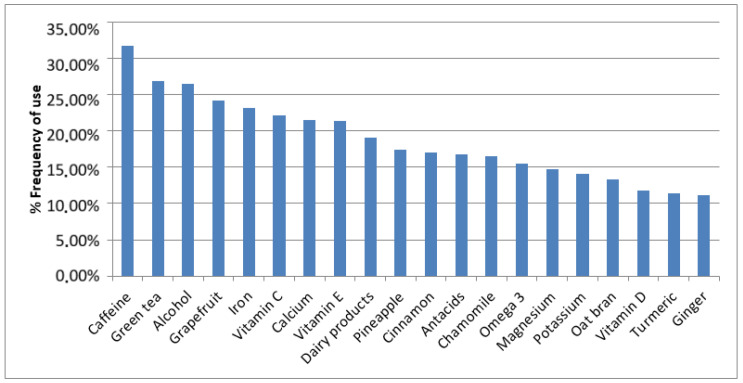
Most frequent lifestyle habits of the general population.

**Figure 4 pharmaceuticals-17-00565-f004:**
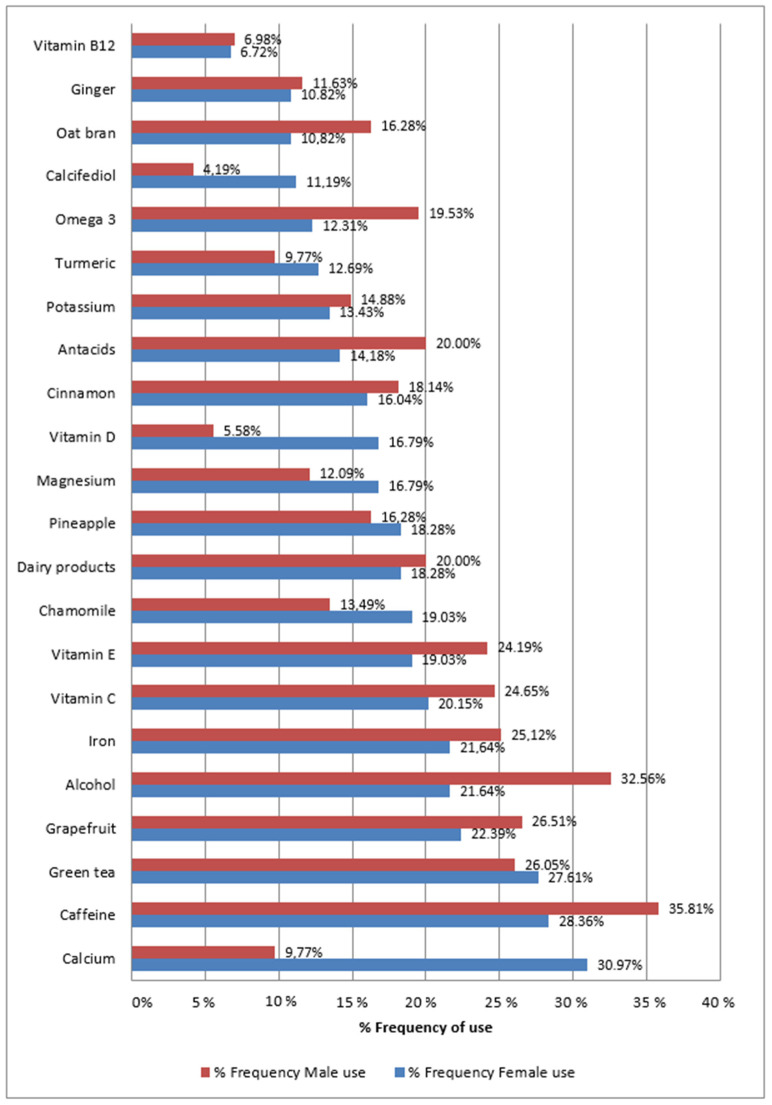
Most frequent active ingredients of lifestyle habits used by female (blue) and male (red) population.

**Figure 5 pharmaceuticals-17-00565-f005:**
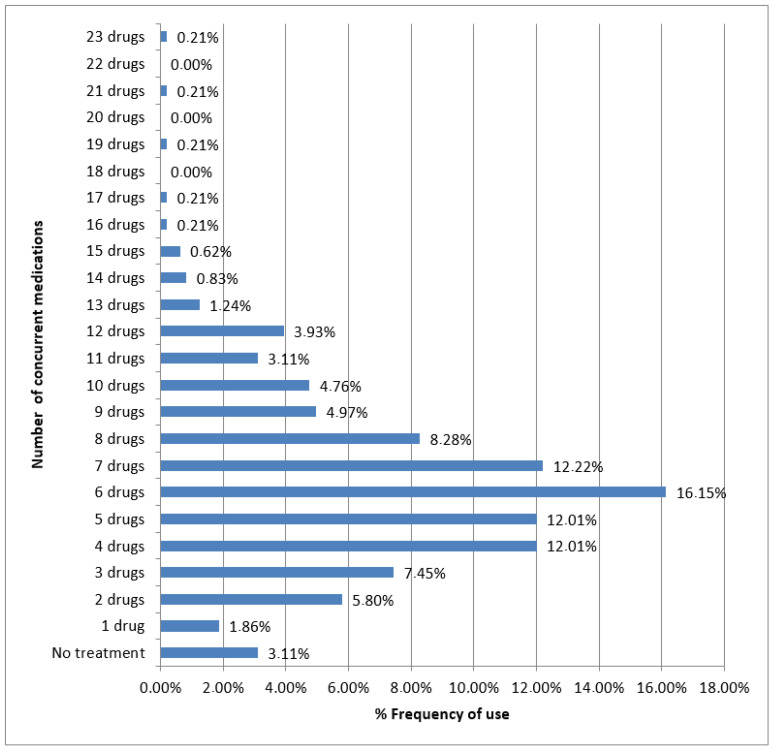
Distribution of the frequency of concurrent medications in the study population.

**Table 1 pharmaceuticals-17-00565-t001:** Most common type of drugs or treatment with percentage of use by gender.

Type of Drug/Treatment	Frequency of Use (%)	Male Frequency of Use (%)	Female Frequency of Use (%)
Antihypertensives	72.26%	78.60%	67.16%
Platelet aggregation inhibitors/anticoagulants	65.84%	68.37%	63.81%
Cholesterol lowering drugs	55.49%	56.74%	54.48%
Gastroprotective agents	52.17%	50.23%	53.73%
Sleep disorder treatment	34.78%	24.19%	43.28%
Diuretics	32.92%	32.56%	33.21%
Analgesics	32.30%	22.33%	40.30%
Anxiolytics	30.85%	20.47%	39.18%
Antihyperglycemics/anti-diabetic agents	27.33%	32.56%	23.13%
Antidepressants	24.02%	17.21%	29.48%
Prostate treatment	17.39%	37.67%	0.00%
Anti-inflammatories	17.18%	14.88%	19.03%
Thyroid hormone therapy	15.73%	7.44%	22.39%
Osteoporosis/osteoarthritis treatment	10.35%	2.79%	16.42%
Anticonvulsants	9.94%	8.37%	11.19%
Gout treatment	8.90%	16.74%	2.61%
Antiarrhythmics	8.70%	7.44%	9.70%
Asthma treatment	8.70%	7.44%	9.70%
Bronchodilators	8.49%	9.30%	7.84%
Vasodilators	8.49%	9.30%	7.84%
Overactive bladder (OAB) treatment	7.45%	11.16%	4.48%
Anti-allergics	6.83%	6.51%	7.09%
Arthritis treatment	6.42%	4.65%	7.84%
Corticosteroids	6.00%	6.98%	5.22%
Glaucoma/ocular hypertension treatment	5.80%	5.58%	5.97%

**Table 2 pharmaceuticals-17-00565-t002:** Potential side effects that may arise from combining these drugs.

Objective Drug	Precipitating Drug	Occurrences	Type of Interaction
Acetylsalicylic Acid	Omeprazole	71	Moderate inhibitor of the enzyme CYP2C9.
	Dipyrone	5	Competes for the binding to COX-1, reducing the antiplatelet effect of aspirin.
	Citalopram	2	May potentiate the inhibition of platelet aggregation caused by aspirin.
Amlodipine	Omeprazole	27	Weak inhibitors of the CYP3A4 enzyme.
	Atorvastatin	18	
	Acetaminophen	14	
	Dipyrone	4	Weak inducer of the gene CYP3A4.
	Acetaminophen	14	Weak inducer of the gene CYP3A5.
	Bisoprolol	13	Can cause atrioventricular conduction disorders, left ventricular failure, and hypotension.
Atorvastatin	Bisoprolol	34	Potent inhibitor of efflux protein Pgp-MDR1, encoded by gene ABCB1.
	Dipyrone	10	Weak inducer of the gene CYP3A4.
Bisoprolol	Omeprazole	37	Weak inhibitors of the CYP3A4 enzyme.
	Acetaminophen	16	
	Atorvastatin	34	
	Dipyrone	5	Weak inducer of the gene CYP3A4.
	Acetaminophen	16	Weak inducer of the gene CYP2D6.
	Amlodipine	13	Can cause atrioventricular conduction disorders, left ventricular failure and hypotension.
Citalopram	Simvastatin	1	Weak inhibitors of efflux transport protein Pgp-MDR1, encoded by gene ABCB1.
	Atorvastatin	2	
	Atorvastatin	2	Weak inhibitors of the CYP3A4 enzyme.
	Omeprazole	2	
	Acetaminophen	3	
	Bisoprolol	1	Potent inhibitor of efflux protein Pgp-MDR1, encoded by gene ABCB.
	Omeprazole	2	Moderate inhibitor of the enzyme CYP2C19.
	Dipyrone	2	Weak inducer of the gene CYP3A4.
	Acetylsalicylic acid	2	Weak inducer of the gene CYP2C19.
	Acetaminophen	3	Weak inducer of the gene CYP2D6.
	Tramadol	2	Can cause serotonin syndrome.
	Hydrochlorothiazide	3	Increases risk of hyponatremia and associated symptoms (confusion, disorientation, weakness).
Dipyrone	Omeprazole	18	Moderate inhibitor of the enzyme CYP2C19 and CYP2C9.
	Acetylsalicylic acid	5	Weak inducer of the gene CYP2C19.
Furosemide	Tramadol	4	Reduction in the efficacy of diuretics because opioids induce the antidiuretic hormone secretion.
	Enalapril	7	May lead to severe hypotension and deterioration in renal function.
Lorazepam	Acetaminophen	10	Weak inductor of the enzyme UGT2B7.
Enalapril	Omeprazole	26	Weak inhibitors of the CYP3A4 enzyme.
	Atorvastatin	14	
	Acetaminophen	10	
	Dipyrone	5	Weak inducer of the gene CYP3A4.
	Valsartan	0	Can cause renal failure, hypotension, and hypokalemia.
	Olmesartan	0	
	Furosemide	7	Can cause precipitous fall in blood pressure in some patients.
	Acetylsalicylic acid	20	Increases the blood pressure by inhibiting the renal synthesis of prostaglandins and antagonizes the effect of enalapril.
Levothyroxine	Omeprazole	26	Reduce the absorption of levothyroxine.
	Pantoprazole	5	
Metformin	Bisoprolol	18	Inhibitor of influx transport protein OCT2, encoded by SLC22A2 gene.
	Olmesartan	4	May increase the effect of metformin and facilitate hypoglycemia.
	Valsartan	8	
	Levothyroxine	12	Can destabilize the control of blood glucose.
	Hydrochlorothiazide	12	May impair the control of blood glucose in diabetic patients.
Olmesartan	Furosemide	6	Can lead to severe hypotension and deterioration in renal function, including renal failure.
	Acetylsalicylic acid	7	Can cause a reduction in the antihypertensive effect.
Omeprazole	Simvastatin	32	Weak inhibitors of the efflux transport protein Pgp-MDR1, encoded by the gene ABCB1.
	Atorvastatin	49	
	Bisoprolol	37	Potent inhibitor of efflux protein Pgp-MDR1, encoded by gene ABCB1.
	Dipyrone	18	Weak inducer of the gene CYP3A4.
	Acetylsalicylic acid	71	Weak inducer of the gene CYP2C19.
Pantoprazole	Simvastatin	7	Weak inhibitors of efflux transport protein Pgp-MDR1, encoded by gene ABCB1.
	Atorvastatin	14	Weak inhibitors of efflux transport protein Pgp-MDR1, encoded by gene ABCB1. Weak inhibitors of the CYP3A4 enzyme.
	Omeprazole	0	Weak inhibitor of the CYP3A4 enzyme.
	Acetaminophen	5	Weak inhibitor of the CYP3A4 enzyme.
	Bisoprolol	10	Potent inhibitor of efflux protein Pgp-MDR1, encoded by gene ABCB1.
	Omeprazole	0	Moderate inhibitor of the enzyme CYP2C19.
	Dipyrone	2	Weak inducer of the gene CYP3A4.
	Acetylsalicylic acid	9	Weak inducer of the gene CYP2C19.
Simvastatin	Omeprazole	32	Weak inhibitors of CYP3A4.
	Atorvastatin	0	
	Acetaminophen	16	
	Dipyrone	4	Weak inducer of the gene CYP3A4.
	Acetaminophen	16	Weak inducer of the enzyme UGT2B7.
	Amlodipine	13	Increases simvastatin blood concentrations, may increase risk of myotoxicity.
Valsartan	Omeprazole	15	Moderate inhibitor of the enzyme CYP2C9.
	Acetylsalicylic acid	14	Reduces the renal function.
	Furosemide	2	Can cause severe hypotension and deterioration in renal function.
Tramadol	Omeprazole	5	Weak inhibitor of influx transport protein OCT1, encoded by gene SLC22A1. Weak inhibitor of the CYP3A4 enzyme.
	Pantoprazole	2	Weak inhibitor of influx transport protein OCT1, encoded by gene SLC22A1.
	Atorvastatin	3	Weak inhibitor of the CYP3A4 enzyme.
	Dipyrone	2	Weak inductor of the CYP3A4 and CYP2B6 genes.
	Acetaminophen	6	Weak inductor of the CYP2D6 gene.
			Weak inhibitors of the CYP3A4 enzyme.
	Lorazepam	3	Can increase hypotension risk, respiratory depression, deep sedation, coma, and death.

**Table 3 pharmaceuticals-17-00565-t003:** Possible side effects resulting from drug–lifestyle product combinations.

Objective Drug	Lifestyle Habit	Type of Interactions
Acetaminophen	Turmeric	Weak inducer of the gene CYP2A6.
	Alcohol	Weak inducer of the gene CYP2E1.
		Can cause higher levels of the compound NADPQ1, which is very hepatotoxic.
Acetylsalicylic Acid	Pineapple	Potent inhibitor of the CYP2C9 enzyme.
	Ginger	Inhibits the thromboxane synthase activity with which can interact with anticoagulants in a significant way.
	Alcohol	Can increase aspirin-induced gastric mucosal damage and aspirin-induced prolongation of the bleeding time.
Allopurinol	Turmeric	Potent inhibitor of the efflux transport protein ABCG2, encoded by the gene BCRP.
Amlodipine	Grapefruit	Grapefruit is a potent inhibitor of the enzyme CYP3A4, increasing the bioavailability of the drug by a factor greater than five causing toxicity due to overdose.
	Chamomile	Moderate inhibitor of the enzyme CYP3A4.
	Green tea	Weak inhibitors of the CYP3A4 enzyme.
	Caffeine	
	Alcohol	Weak inducer of the gene CYP3A4.
Atorvastatin	Cinnamon	Regular cinnamon intake can lead to an exposure to one of its compounds, coumarin. This may have hepatotoxic effects that could cause hepatitis when combine with statins.
	Oat bran	Decrease the atorvastatin pharmacological effect.
	Chamomile	Moderate inhibitor of the enzyme CYP3A4.
	Grapefruit	Potent inhibitor of the enzyme CYP3A4.
		Potent inhibitor of the efflux transport protein Pgp-MDR1, encoded by the gene ABCB1.
		Potent inhibitor of the influx carrier protein OATP1B1, encoded by the gene SLCO1B1.
	Turmeric	Potent inhibitor of the efflux transport protein ABCG2, encoded by the gene BCRP.
		Potent inhibitor of the efflux transport protein Pgp-MDR1, encoded by the gene ABCB1.
	Alcohol	Weak inducer of the gene CYP3A4.
Bisoprolol	Grapefruit	Potent inhibitor of the enzyme CYP3A4.
	Chamomile	Moderate inhibitor of the enzyme CYP3A4.
	Green tea	Weak inhibitors of the CYP3A4 enzyme.
	Caffeine	
	Alcohol	Weak inducer of the gene CYP3A4.
Citalopram	Green tea	Weak inhibitor of the efflux transport protein Pgp-MDR1, encoded by the gene ABCB1.
	Grapefruit	Potent inhibitor of the enzyme CYP3A4.
		Potent inhibitor of the efflux transport protein Pgp-MDR1, encoded by the gene ABCB1.
	Chamomile	Moderate inhibitor of the enzyme CYP3A4.
	Green tea	Weak inhibitors of the CYP3A4 enzyme.
	Caffeine	
	Turmeric	Potent inhibitor of the efflux transport protein Pgp-MDR1, encoded by the gene ABCB1.
	Alcohol	Weak inducer of the gene CYP3A4.
Dipyrone	Pineapple	Potent inhibitor of the CYP2C9 enzyme.
Enalapril	Grapefruit	Potent inhibitor of the enzyme CYP3A4.
	Chamomile	Moderate inhibitor of the enzyme CYP3A4.
	Green tea	Weak inhibitors of the CYP3A4 enzyme.
	Caffeine	
	Alcohol	Weak inducer of the gene CYP3A4.
	Potassium	Can lead to a potassium retention that can cause hyperkalemia.
Hydrochlorothiazide	Calcium	Increase the risk of hypercalcemia.
	Alcohol	Potentiates the appearance of orthostatic hypotension.
Levothyroxine	Magnesium	Can reduce levothyroxine bioavailability; some patients may develop hypothyroidism.
	Antacids	
	Calcium	Reduces the absorption of the drug by approximately 33%.
	Caffeine	Limited clinical evidence suggest that ingestion of coffee may reduce the drug bioavailability
	Iron	Could reduce the drug bioavailability.
Lorazepam	Alcohol	Increases the hypotension risk, respiratory depression, deep sedation, coma, and death
Metformin	Green tea	Inhibitor of the influx transport protein OCT2 in the basolateral membrane of the renal proximal tubule, encoded by the SLC22A2 gene; there will be less intestinal absorption of the drug causing a lower bioavailability and possible therapeutic failure.
	Ginger	Can increase insulin levels and/or lower blood glucose levels that could lead to hypoglycemia.
	Alcohol	Could potentiate the risk of lactic acidosis.
Olmesartan	Potassium	May lead to increases in potassium in serum.
Omeprazole	Pineapple	Potent inhibitor of the CYP2C9 enzyme.
	Green tea	Weak inhibitors of the efflux transport protein Pgp-MDR1, encoded by the gene ABCB1.
	Grapefruit	Potent inhibitor of the enzyme CYP3A4.
		Potent inhibitor of the efflux transport protein Pgp-MDR1, encoded by the gene ABCB1.
	Chamomile	Moderate inhibitor of the enzyme CYP3A4.
	Turmeric	Potent inhibitor of the efflux transport protein Pgp-MDR1, encoded by the gene ABCB1.
	Alcohol	Weak inducer of the gene CYP3A4.
Pantoprazole	Green tea	Weak inhibitors of the efflux transport protein Pgp-MDR1, encoded by the gene ABCB1.
	Grapefruit	Potent inhibitor of the enzyme CYP3A4.
		Potent inhibitor of the efflux transport protein Pgp-MDR1, encoded by the gene ABCB1.
	Chamomile	Moderate inhibitor of the enzyme CYP3A4.
	Green tea	Weak inhibitors of the CYP3A4 enzyme.
	Caffeine	
	Turmeric	Potent inhibitor of the efflux transport protein Pgp-MDR1, encoded by the gene ABCB1.
	Alcohol	Weak inducer of the gene CYP3A4.
Simvastatin	Cinnamon	Regular cinnamon intake can lead to an exposure to one of its compounds, coumarin. This may have hepatotoxic effects that could cause hepatitis when combine with statins.
	Oat bran	Decreased simvastatin pharmacological effect.
	Grapefruit	Potent inhibitor of the enzyme CYP3A4.
		Potent inhibitor of the influx carrier protein OATP1B1, encoded by the gene SLCO1B1.
	Chamomile	Moderate inhibitor of the enzyme CYP3A4.
	Green tea	Weak inhibitors of the CYP3A4 enzyme.
	Caffeine	
	Turmeric	Potent inhibitor of the efflux transport protein ABCG2, encoded by the gene BCRP.
	Alcohol	Weak inducer of the gene CYP3A4.
Valsartan	Pineapple	Potent inhibitor of the CYP2C9 enzyme.
	Grapefruit	Potent inhibitor of the influx carrier protein OATP1B1, encoded by the gene SLCO1B1.
	Potassium	May lead to increases in potassium in serum.
Tramadol	Grapefruit	Potent inhibitor of the enzyme CYP3A4.
	Chamomile	Moderate inhibitor of the enzyme CYP3A4.
	Green tea	Weak inhibitors of the CYP3A4 enzyme.
	Caffeine	
	Alcohol	Weak inducer of the gene CYP3A4.

**Table 4 pharmaceuticals-17-00565-t004:** Genes linked to the metabolism of most used drugs.

Frequent Drug	Evidence	Gene
Acetaminophen	4	UGT1A9
Acetylsalicylic acid	3	CYP2C9
Allopurinol	1	HLA-B5801
Amlodipine	3	CYP3A4
Atorvastatin	3	BCRP
	2	CYP3A4
	2	SLCO1B1
Bisoprolol	4	CYP3A4
Citalopram	1	ABCB1
	2	CYP2C19
Dypirone	4	CYP2C19
	1	G6PD
		NAT2
Enalapril	1	G6PD
Lorazepam		UGTB7
Omeprazole	3	CYP2C19
Simvastatin	4	BCRP
	2	CYP3A4
	1	SLCO1B1
Tramadol	1	CYP2D6
	4	CYP3A4
	2	SLC22A1
Valsartan	4	SLCO1B1
Hydrochlorothiazide	3 (PharmGKB)	PRKCA
	3 (PharmGKB)	NEDD4L
	3 (PharmGKB)	YEATS4
Pantoprazole	1A (PharmGKB)	CYP2C19
Furosemide	3 (PharmGKB)	ADD1

## Data Availability

All supporting data reported in this study are included in the manuscript.

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
