# Peer review of "Comprehensive Analysis of Drug Utilization Patterns, Gender Disparities, Lifestyle Influences, and Genetic Factors: Insights from Elderly Cohort Using g-Nomic® Software"

_pharmaceuticals, 2024, doi:10.3390/ph17050565_

Round 1

Reviewer 1 Report

Comments and Suggestions for Authors

Rodríguez Castillo et al. analysed the pharmacological data from 483 patients, primarily elderly and polymedicated, by using Eugenomic®'s personalized prescription software, g-Nomic®.

Study design is not entirely clear. In my opinion, English language should be reviewed, and extensive editing required for typos, linguistic or punctuation inaccuracies.

The description of the “materials and methods” and “results” is approximate and superficial, the main objectives of the study are not specified, and the discussion does not show an adequate dissertation of results.

Furthermore, to make this article suitable for publication, some corrections are mandatory:

1)    The title of the manuscript does not reflect the aim/main findings of the study. Please edit it.

2)    Please specify the main objectives of the study in the abstract and in the “introduction” section.

3)   Authors state that “patient consent was waived due to being a retrospective anonymized study in which we only obtained cases and medication information”. Were patients never informed about the possible use of their clinical/pharmacological data for research purposes? Please, clarify.

4)    Please review all figure and table legends. They are sparse and non-explanatory. Use Arabic numerals or Roman numerals. Fix typos, such as replacing “frecuency” with “frequency” in figure 1.

5)    Why are the interactions between furosemide and lifestyle habits missing in Table 3?

6)    Why are the drugs represented in Figure 2 different from those in figure 1 (e.g acenocoumarol is not listed in Figure 1 despite its high frequency in the male and female cohort)? Please explain better.

7)    Please check numbers of paragraph and subparagraph.

8) Table 4 shows the most relevant genes associated with the metabolism of the drugs analysed and their level of evidence, based on clinical annotation according to established guidelines, technical sheets, and relevant scientific publications. For example, why is the SLCO1B1 gene missing for simvastatin despite its high level of evidence (1A) on the PharmGkb guidelines? Why were only four genes chosen for hydroclorotiazide and furosemide? Please, check all data reported in the table and whether the g-Nomic software shows the same pharmacogenes and their level of evidence reported in the PharmGKB and CPIC guidelines.

9)    In lines 394-396, authors express that “Physicians then questioned the patients about the use of these active ingredients, and if confirmed, the ingredient was added to the list of consumed lifestyle habits…”. Thus, was the information on lifestyle habits entered into the database later? How can this be a retrospective study? Please, clarify.

10) In the “study population” section (materials and methods) data about pharmacological therapies (numbers of concurrent medications) should not be represented in the figure and included in this section. They could be in the “Results” section.

11) In line 300 “most of the study population (91.30%) falls into the elderly category, aged 65 years or older”, while in lines 364-365 it is reported “A significant proportion of the study population (91.30%) with ages ranging from 40 to 92 years old”. What is the age range of the study population?

Finally, in “Conclusions” it may also be interesting to mention any limitations of this study.

Reviewer 2 Report

Comments and Suggestions for Authors

This paper reports on a study conducted among 483 patients undergoing pharmacological treatments provided by a health resort located in the province of Valencia, Spain, investigating the medications and ‘lifestyle’ they were taking, which was manually inputted into the g-Nomic® personalized prescription software, that overviewed molecular interactions between selected drugs, putatively allowing observation of all potential side effects associated with specific drug combinations, as reported by g-Nomic®. The associated genes involved in the metabolism of each drug were examined in the "Genes" section of g-Nomic® - for drugs without genetic information provided by the software, a literature search was conducted using the Clinical Pharmacogenetics Implementation Consortium (CPIC) and the Pharmacogenomics Knowledgebase (PharmGKB). The authors found that the most prescribed drug classes included antihypertensives, platelet aggregation inhibitors, cholesterol-lowering drugs, and gastroprotective medications. Drug-lifestyle interactions primarily involved inhibitions but also included inductions. Interactions were analyzed considering gender. Significant genetic variants identified in the study encompassed ABCB1, SLCO1B1, CYP2C19, CYP2C9, CYP2D6, CYP3A4, ABCG2, NAT2, SLC22A1, and G6PD. The authors strongly recommended to consider pharmacogenetics testing to prevent adverse reactions and enhance medication effectiveness.

The authors may wish to address the following:

1.      Line 181, 403, 410, etc - potential interactions was addressed only between the 20 most frequently appearing drugs – it may make more clinical value to instead investigate interactions that could have serious health effects on the patient

2.      Lines 183-184 - ‘potential side effects that may arise from combining these drugs’ – were these side effects really seen in the patients studied? If not, what is the clinical value of this testing?

3.      Table 2 – is it correct to assume that interactions can occur not matter if a drug in pair was objective or precipitating? If yes, why is it eg atorvastatin appears in the amlodipine list but not vice-versa?

4.      Line 231, 339 - ‘recommended to perform pharmacogenetic testing to prevent ADRs’ - were the gene status actually tested in the patients?

5.      Line 390-395 – ‘lifestyle’ was asked only after the drugs information was entered into the software – it should have been asked a-priori when the dug information was being collected, to avoid bias

6.      Conflicts of interest – I note that 2 of the 4 authors work in Eugenomics, which developed g-Nomic®

Reviewer 3 Report

Comments and Suggestions for Authors

The manuscript presents the results of the pharmacogenetic analysis of the medical prescriptions of a cohort of elderly patients using the g-Nomic® software.

The manuscript is well-written and organized and presents relevant results for the complex field of drug-drug interaction management.

However, I have a few comments for the authors:

  • In the Introduction, the authors should indicate whether other papers present results based on this software.
  • The authors must outline the main objective of their work and the primary outcome that emerged from the analyzed prescription dataset.
  • The authors should present the frequency (%) of the drug-drug interactions presented in Tables II(a) and II(b) in section 2.4. 
  • I have the same observation for section 2.5, Tables III(a-c).
  • Please revise the numbering of the sections (e.g., there are two 2.5 sections).
  • In the section Genes and Haplotypes Associated with the Metabolism of the Most Frequent Drugs in the Study Population, the authors should provide more details about selecting evidence from clinical reports and establishing the evidence strength score. It would be relevant for the readers to understand this procedure better if the authors provided 2 to 3 examples of drug and gene(s) collecting and scoring evidence.
  • The authors must add the most relevant conclusion(s) after analyzing their prescription dataset using g-Nomic software.

Reviewer 4 Report

Comments and Suggestions for Authors

The manuscript describes the results of the analysis of the numerous drugs administered by patients between 40 and 92 years old obtained using Eugenomic®'s personalized prescription software, g-Nomic®. My major comment concerns the aim of the study, because as it is not clearly defined in the Introduction, it is very difficult to assess the content of the manuscript as well as the conclusions drawn. My comments are listed below:

·        The Abstract does not include any results.

·        Line 35: Are the authors sure that the one drug participating in the interaction is called ‘precipitating drug’?

·        Line 50: What is the difference between drug level and concentration?

·        I did not find the aim of the study in the Introduction.

·        Paragraph 2.3: The terms ‘Active Ingredients’ and ‘lifestyle habits’ should be clearly defined. Are these only food ingredients or also some supplements or even drugs? In Figure 3, there are vitamins (E, C), antacids and food ingredients or spices (oat bran, gingered) mentioned as they were of one category.

·        Most of the Results repeat data from figures (lines 129-140, 165-176). In my opinion, it should be corrected.

·        The captions of Figures 2-5 as well as Tables 2-4 are to short and not informative. Moreover, Tables should be numbered with Arabic numerals. I suggest writing ‘continuation’ for the second part of Table 2 instead of dividing Table 2 to Table 2a and 2b. The same applies to Table 3.

·         In my opinion, the age range of the studied population is too large (40-92 years). It should be divided into more narrow ranges as the activity of forty years old is much different than of sixty years old. Due to the large age range of patients, the results are less informative and do not lead to any practical and reliable conclusions. Moreover, calling forty-years-old ‘an elderly person’ is rather exaggeration.

·        Lines 328-336: These sentences do not discuss the results with the literature data but provide some well-known theoretical data.

·        The conclusions section is rather too long. It should include the conclusions from the study, not repeat the general information.

Comments on the Quality of English Language

·        Minor language correction is needed.

Round 2

Reviewer 2 Report

Comments and Suggestions for Authors

This is a revised submission of a paper reporting on a study conducted among 483 patients undergoing pharmacological treatments provided by a health resort located in the province of Valencia, Spain, investigating the medications and ‘lifestyle’ they were taking, using the g-Nomic® personalized prescription software.

The authors have responded to my queries

The authors may wish to address the following:

1.      That the interactions of only the 20 most frequently appearing drugs was studied should be entered as a study limitation

2.      That there was no data on the actual occurence of the potential side effects should be entered as a study limitation, and this can be mentioned as possible follow-up study

3.      Table 2 – I feel that the drug interactions should be mentioned whether objective or precipitant, with the explanatory text. This is because the list is long and readers may only look at the first column for drugs they are interested in Thus, in this example, both atorvastatin and amlodipine should be in column 1 and mention the other

Author Response

1.      That the interactions of only the 20 most frequently appearing drugs was studied should be entered as a study limitation
It should be noted that the study's focus on only the interactions of the 20 most frequently prescribed drugs constitutes a limitation. Although the studied cohort encompassed a total of 330 active ingredients, considering all possible combinations would result in a vast number of permutations, estimated at approximately 54,285 pairs. Consequently, for practical reasons, we narrowed our analysis to the combinations involving the top 20 most prescribed drugs, accounting for their respective frequencies. Therefore, the limitation lies in the restricted scope of exploring all potential drug interactions and the actual ocurrence of the potential side effects. occurrences.

2.      That there was no data on the actual occurence of the potential side effects should be entered as a study limitation, and this can be mentioned as possible follow-up study.

This issue has been included in the response to the first comment of this very same reviewer

3.      Table 2 – I feel that the drug interactions should be mentioned whether objective or precipitant, with the explanatory text. This is because the list is long and readers may only look at the first column for drugs they are interested in Thus, in this example, both atorvastatin and amlodipine should be in column 1 and mention the other.

There is no difference between objective and precipitating drug, it was a nomenclature used just to denote drug interactions.  It is the same that appears in either column. Only the most described interactions influencing the metabolism of the first drug was displayed in the second column.

Reviewer 3 Report

Comments and Suggestions for Authors

The authors have adequately addressed almost all comments and inserted them consistently in the revised manuscript.

However, a minor observation remains: The initial comment to assign percentages to potential drug interactions started from the patient cohort dataset analyzed in this manuscript. Authors should possess data on whether objective drugs were combined with precipitating drugs and be able to provide results on the incidence of these associations. Therefore, authors should either add the percentages of these drug associations in the tables or bring some clarifications in the main text.

Author Response

However, a minor observation remains: The initial comment to assign percentages to potential drug interactions started from the patient cohort dataset analyzed in this manuscript. Authors should possess data on whether objective drugs were combined with precipitating drugs and be able to provide results on the incidence of these associations. Therefore, authors should either add the percentages of these drug associations in the tables or bring some clarifications in the main text.

For us there is no difference between objective and precipitating drug. It is a matter of nomemclature to indicate an interaction between 2 drugs. Regarding the number of interactions of between each objective and precipitating drug, they are shown now in the revised version of the manuscript to provide the incidence of these associations.

Reviewer 4 Report

Comments and Suggestions for Authors

All my comments were correctly addressed.

Author Response

Thank you.

We attach here the last reviewd version
